materials science/synthetic chemistry

aerosol-assisted method, mesoporous silicate molecular sieves, sustained-release drug carriers

**Author for correspondence:**
Yang Wang
e-mail: 563484708@qq.com

This article has been edited by the Royal Society of Chemistry, including the commissioning, peer review process and editorial aspects up to the point of acceptance.

# Synthesis of mesoporous silicate molecular sieves by the aerosol-assisted method for loading and release of drug

## Liang Hao, Xiaojia Li and Yang Wang

School of Fundamental Sciences, China Medical University, Shenyang 110122, People's Republic of China

LH, 0000-0001-8449-4869

The mesoporous silicate molecular sieves were synthesized with polyether F127 as the template by the aerosol-assisted method for loading and release of ibuprofen (IBU). The synthesized samples were characterized by scanning electron microscopy, transmission electron microscopy, X-ray diffraction and $N_2$ adsorption–desorption isotherms. The drug IBU was applied as a model drug to investigate the drug release behaviour by ultraviolet spectrophotometry measurements. The investigation results demonstrate that mesoporous silicate molecular sieves by the aerosol-assisted method are spherical with a core–shell structure. As the drug carrier, it has good structural stability and can achieve drug controlled release which is expected. It exhibits safety to a certain degree. Therefore, the aerosol-assisted synthesis method provides a new idea for the synthesis of sustained-release drug carriers.

## 1. Introduction

In recent years, the research of drug-sustained-release preparations has been a hot topic in the medical community [1–3]. Drug-sustained-release preparations have good application value in maintaining a stable blood concentration, reducing toxic and side effects, prolonging the time of drug effect, and facilitating patient convenience [4]. The design and development of sustained-release drug carriers is one of the important research topics [5]. At present, carriers of traditional drug-sustained-release preparations are materials such as polymer compounds, liposomes, organic acids, polysaccharides and cellulose. The particle size of these carriers is generally above the micron level, and there are many problems such as excessive volume and uneven drug loading. In particular, polymer materials have poor biodegradability, and there is a

hidden danger of environmental pollution. In addition, the use of cross-linking agents in the preparation process will directly affect the efficacy of the medicine. Therefore, the development of new drug carriers has become an urgent problem in the field of drug-sustained release.

Mesoporous molecular sieve refers to a three-dimensional porous material with a pore size of 2–50 nm. It is mainly characterized by the use of surfactant molecular aggregates as a mesoporous template, and the formation of pore structure in the molecular sieve can be achieved through the interface assembly process of surfactant molecular aggregates and inorganic species. Its structure and performance are between the amorphous inorganic aluminosilicate porous material and the inorganic zeolite molecular sieve porous material with a crystalline structure. It not only has a regular pore structure and an amorphous pore wall, but also has some excellent properties which are not available in other porous materials [6]. Since the preparation of mesoporous molecular sieves by the United States Mobile Corporation in 1992, owing to their orderly and controllable pore structure, huge specific surface area, non-pharmacological activity and non-toxicity, they has demonstrated great application potential and unparalleled superiority in separation, catalysis, biomedicine and drug delivery [7–9]. Professor Vallet Regi's research team first used mesoporous molecular sieve MCM-41 as a drug carrier in 2001 and loaded ibuprofen (IBU) [10]. The results of the study found that using MCM-41 as a slow-release carrier for IBU not only overcame the disadvantages of the traditional slow-release matrix such as polymers and the uneven mixing of the drug, but also that the system can prolong the release cycle of the drug. In recent years, research on the use of mesoporous molecular sieves as drug carriers has increased significantly [10–12]. A series of mesoporous molecular sieves, such as mesoporus crystalline material-48 (MCM-48), Santa Barbara amorphous material-15 (SBA-15), Michigan State University (MSU), mesoporous (siliceous) oxides (TUD), and hexagonal mesoporous silica (HMS) have been reported to be used as drug carriers and controlled-release systems [12–18]. In addition, by modifying mesoporous silica as a drug carrier, it has shown great application in drug delivery, controlled release, tumour-targeted therapy and reversal of drug resistance [19–23].

There are three main methods for the synthesis of mesoporous molecular sieves: the sol–gel method, the hydrothermal synthesis method and the phase transition method [24–27]. However, there are some problems with these methods, such as low product yield, a large amount of template agent and high production cost and serious wastewater pollution [28–30]. Aerosol-assisted synthesis is expected to solve these problems in traditional methods. The synthesis process is as follows: silicon source, mesoporous template agent, water and ethanol are mixed to form the solution, the aerosol spray is formed by an aerosol generator, and dried by a tube heating furnace [31–36]. The pore template agent forms micelles and self-assembles with the silicon species to form the mesoporous structure. The formed powder is collected by a collector. The entire spray drying process of the molecular sieve product takes only a few seconds. Compared with traditional methods, the aerosol-assisted method shows its unique advantages in the synthesis of molecular sieves: (i) it is easy to control the composition of the product, the size and morphology of the particles; (ii) it has high atom utilization rate and high product yield; (iii) heteroatom distribution is more uniform; (iv) the synthesis process is fast and efficient, and it can effectively reduce water pollution; (v) the structure of the synthesized material is changeable and easy to control; and (vi) the aerosol process can obtain some metastable products, which can be used for products that are difficult to synthesize by water and heat [37–42]. However, there are few reports on the study of the sustained-release and controlled-release effects of aerosol-assisted synthesis of drug carriers.

Herein, we synthesized the novel mesoporous silicate molecular sieves with polyether F127 as a template by the aerosol-assisted method. The morphology of most synthesized samples is spherical with a particle size of about 200 nm. Meanwhile, the well-developed spherical particles have a core–shell structure. The layer of annular pores is arranged in parallel on the outside of the spherical particles, and the three-dimensional short worm-like pore structure is arranged inside the spherical particles. The drug IBU was loaded in the sample as a drug model to study the loading and release of a drug. With the loading of the drug, although the ordering of the carrier is affected to some extent, it still maintains the original structure. At the same time, it can quickly reach the adsorption equilibrium and have a significant slow-release effect during the drug loading process.

# 2. Results and discussion

## 2.1. Preparation of mesoporous silicate molecular sieves by the aerosol-assisted method and supported ibuprofen

The process of aerosol synthesis of molecular sieves is shown in figure 1: the carrier gas atomizes the precursor solution through the nozzle, and excessively large droplets cannot be ejected and flow back

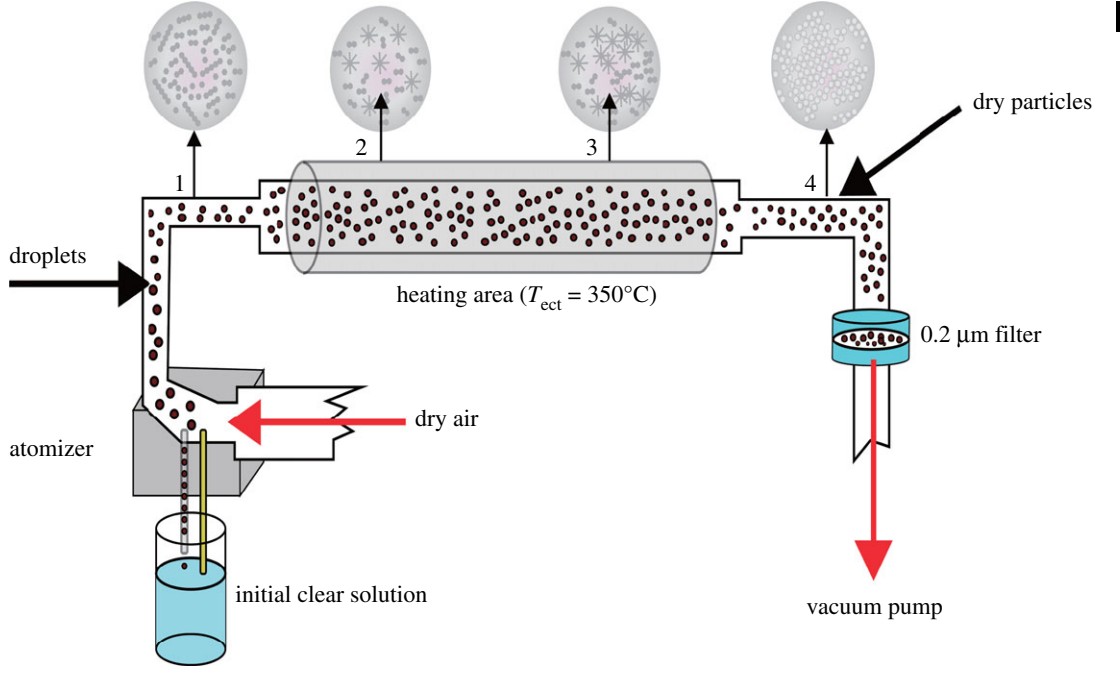

**Figure 1.** The process of aerosol synthesis of molecular sieves.

into the precursor solution. The small droplets are ejected with the carrier gas and the aerosol is formed. Under the impetus of the carrier gas, the aerosol passes through the quartz tube in the tube heating furnace, and the evaporation of the solvent in the quartz tube induces the self-assembling non-volatile matter into the molecular sieve.

The IBU was added into an appropriate amount of ethanol. The mix was magnetically stirred until dissolved. An amount of molecular sieves were added and stirred continuously. After the loading process was completed, the sample was filtered by vacuum suction filtration, washed by ethanol and dried at room temperature to obtain a molecular sieve loaded with IBU.

## 2.2. Scanning electron microscopy

Figure 2 shows scanning electron microscopy (SEM) images of mesoporous silicate molecular sieves by the aerosol-assisted method. The morphology of the synthesized samples is spherical, but the size distribution is different. The particle size of the samples synthesized with polyether F127 as a template is relatively small, most of the particle sizes are below 1 µm, and there are many particles with a particle size of about 200 nm. This is owing to physical properties, such as the viscosity of the mixed solution, resulting in different size distributions of liquid particles formed by spraying.

## 2.3. Transmission electron microscopy

The transmission electron microscopy (TEM) images reveal the well-developed spherical particles with a core–shell structure (figure 3). It can be seen from the figure that the particle size is consistent with the SEM photograph. For the molecular sieves synthesized with polyether F127 as a template agent, the layer of annular pores is arranged in parallel on the outside of the spherical particles, and the three-dimensional short worm-like pore structure is arranged inside the spherical particles. The internal three-dimensional short worm-like channel structure is similar to the KIT-1 TEM photo reported in Ryoo *et al.* [14]. This structure can help molecular sieves become excellent carriers of slow-release drugs.

## 2.4. X-ray diffraction

X-ray diffraction (XRD) is one of the most important means to characterize the structure of molecular sieves. XRD spectra can be used to analyse the sample structure, channel order and other information. As figure 4 shows, the two samples show characteristic peaks (100) and (200) at small angles of 0.80° and 1.5°, respectively, which belong to the characteristic peaks of the mesoporous structure. This shows

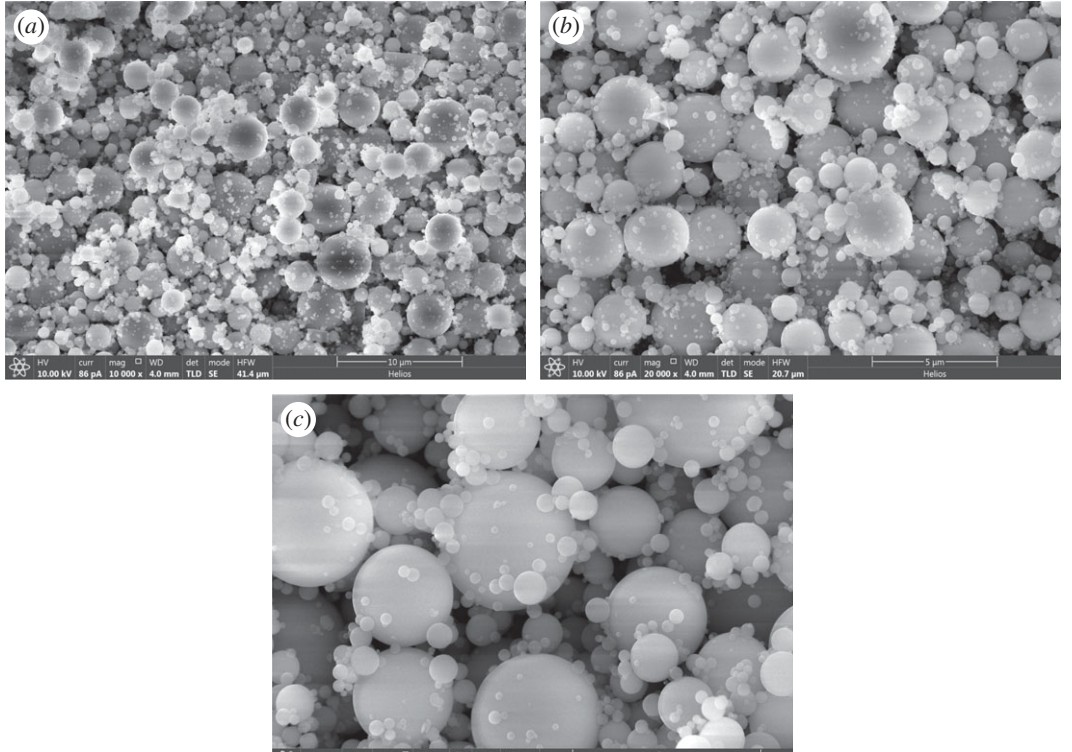

**Figure 2.** SEM of mesoporous silicate molecular sieves by the aerosol-assisted method. (*a,b,c* are different magnifications).

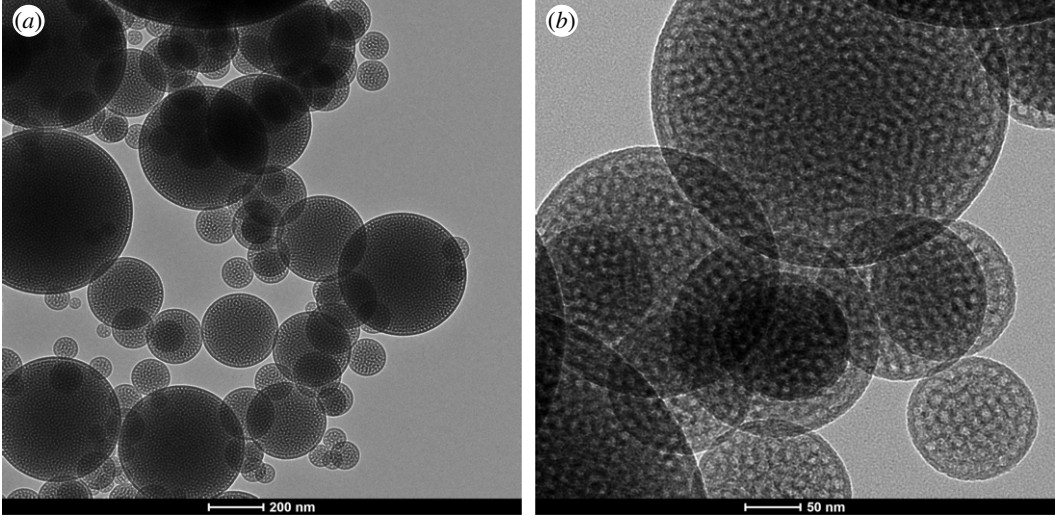

**Figure 3.** TEM of mesoporous silicate molecular sieves by aerosol-assisted method. (*a,b* are different magnifications).

that the molecular sieve carrier by the aerosol-assisted method retained the original pore structure after loading the drug and has good structural stability. However, the intensity of the (100) characteristic peak of the IBU/F127 sample is significantly smaller than that of the F127 sample, indicating that the process of loading the drug caused some damage to the order of the channel structure. In addition, the position of the characteristic peak is related to the pore size. The smaller the $2\theta$ value, the larger the mesoporous structure pore diameter. As can be seen from the figure, the (100) characteristic peak of the IBU/F127 sample shifts significantly to the right, so the pore size of the carrier is reduced.

## 2.5. $N_2$ adsorption–desorption isotherm

The $N_2$ adsorption–desorption isotherms of the mesoporous silicate molecular sieves by the aerosol-assisted method and that supported IBU are shown in figure 5*a*. Both samples have the characteristics

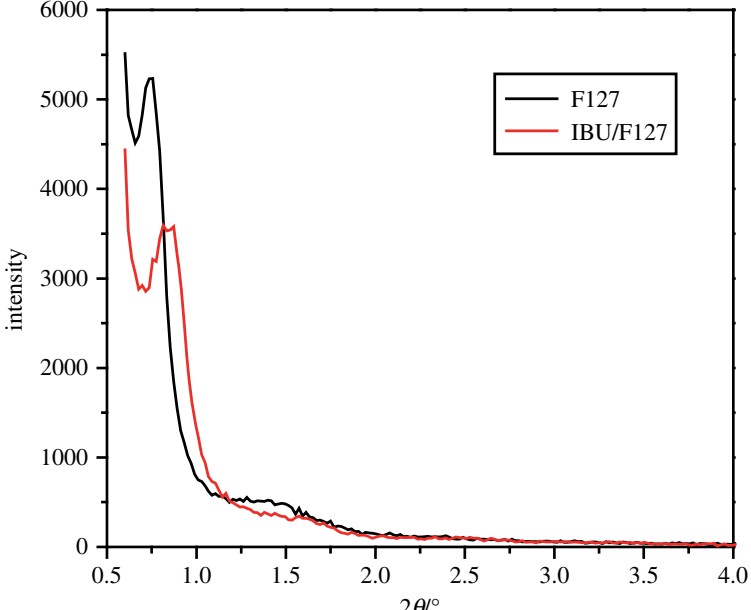

**Figure 4.** XRD of mesoporous silicate molecular sieves by the aerosol-assisted method and that supported ibuprofen.

of type IV and type I isothermal curves, which indicates that the pore structure of the molecular sieve carrier has maintained the original appearance, and basically no pore blocking and collapse have occurred. From the changes of the adsorption amount of these samples, single-layer adsorption occurred in the low-specific pressure region $P/P_0 < 0.01$, and the jump in the $N_2$ adsorption amount is attributed to the filling of micropores, which indicates that the molecular sieve carrier prepared by the aerosol method has a small number of microporous structural units. The intensity of the isotherm of the IBU/F127 sample is very low, and the adsorption capacity is small, which indicates that the sample's pore structure is affected by the loading of the drug. These conclusions are also confirmed in figure 5b. Mesoporous pore size of samples decreased from 8.5 nm to 6.9 nm owing to the loading of drug. The specific surface area of the molecular sieve prepared by the aerosol method is 371.2277 $m^2 g^{-1}$, and the pore volume is 0.3186 $cm^3 g^{-1}$. With the loading of the drug, the specific surface area and pore volume of the sample gradually decrease. This is because after loading the drug, IBU is deposited in the pore channel, which reduces the pore size and pore volume. The specific surface area of IBU/F127 is decreased to 285.1357 $m^2 g^{-1}$, and the pore volume is decreased to 0.2351 $cm^3 g^{-1}$.

## 2.6. The loading of ibuprofen on molecular sieves

The molecular sieve prepared by the aerosol method was as the carrier, and the concentration of IBU was 35 mg $g^{-1}$. The residual drug concentration in the solution was determined by the ultraviolet method, and the loading of IBU on the molecular sieve was calculated by the subtraction method. The loading curve is shown in figure 6. It can be seen that the drug loading process basically reaches equilibrium after 2 h, which is a fast completion process. IBU and the surface of the carrier are bound by non-covalent bonds, and the adsorption equilibrium is reached quickly, which indicates that the molecular sieve prepared by the aerosol method is very suitable as a drug carrier.

## 2.7. The release of ibuprofen on molecular sieves

The amount of IBU released per unit molecular sieve was measured by the ultraviolet method. Figure 7 shows a comparison of the release rate of pure IBU tablets and drug-loaded sample IBU/F127 tablets in phosphate-buffered solution. It can be seen from the figure that the original drug is released completely within 10 min, and the release rate of IBU/F127 reaches 35% in 2 h, indicating that IBU/F127 does have a certain sustained-release effect. This is owing in part to the non-covalent bonding of IBU to hydroxyl groups on the support surface. On the other hand, the worm-like pore structure inside the carrier makes the drug release slowly.

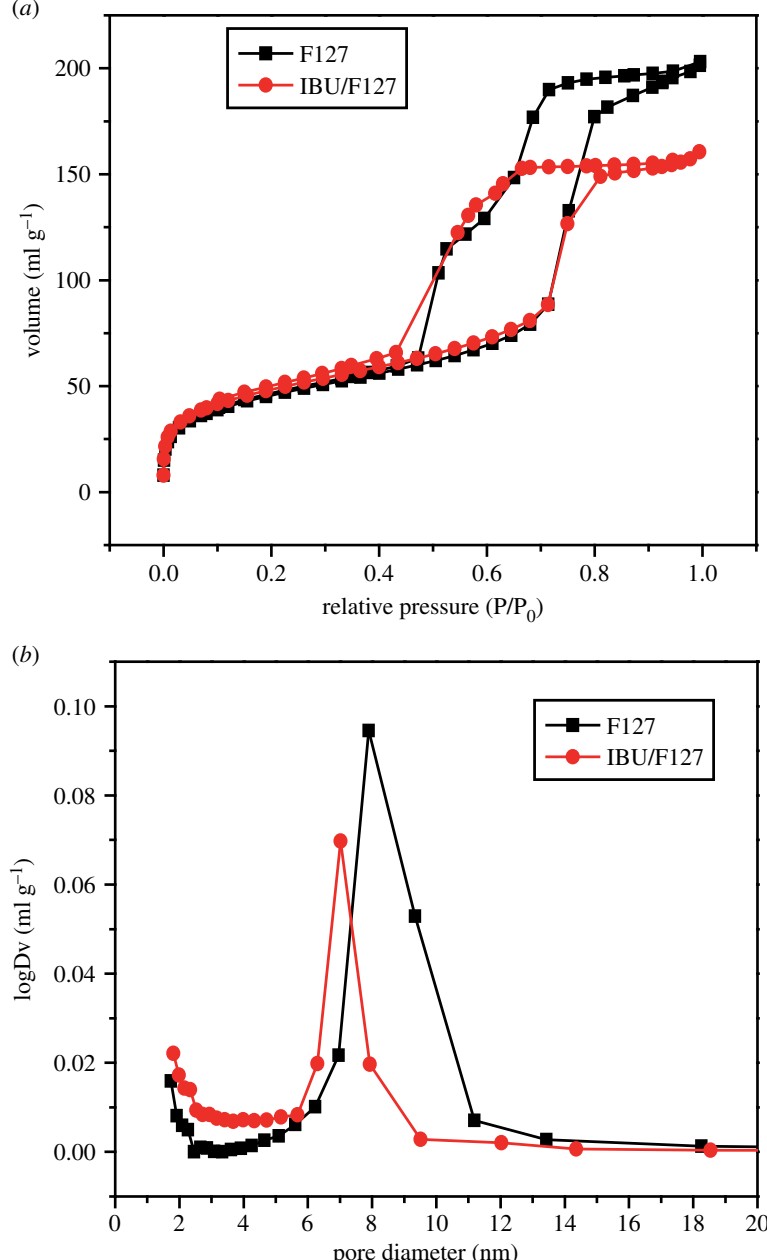

**Figure 5.** $N_2$ adsorption–desorption isotherm of mesoporous silicate molecular sieves by the aerosol-assisted method and that supported ibuprofen. ((*a*) $N_2$ adsorption-desorption isotherms, (*b*) mesopore size distributions).

## 2.8. The bio-safety of materials

The bio-safety of materials was measured by the Cell Counting Kit-8 (CCK-8) assay. As shown in figure 8, the material concentration ranged from 0.5 to 1 µg ml$^{-1}$, and over 80% cell viability could be observed in the HIEC-6 cells incubated with IBU or IBU/F127 for 48 h, exhibiting safety to a certain degree.

# 3. Experimental

## 3.1. Preparation of mesoporous silicate molecular sieves

Tetraethyl orthosilicate (TEOS) was used as the silicon source, and polyvinyl ether-polypropylene ether-polyvinyl ether triblock copolymer (F127) was used as template agents, respectively. The raw material composition is TEOS: F127: ETOH: H2O: HCl = 1 : 0.005 : 18 : 24 : 0.075. The deionized water, ethanol (ETOH), template (F127) and concentrated hydrochloric acid were added to the conical flask and

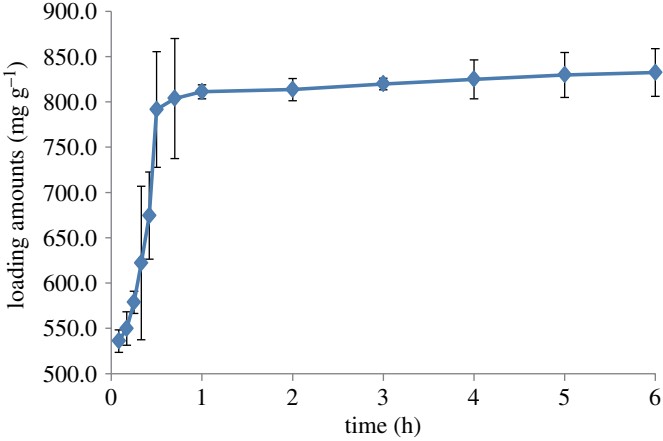

**Figure 6.** Loading curve of ibuprofen on mesoporous silicate molecular sieves by aerosol-assisted method.

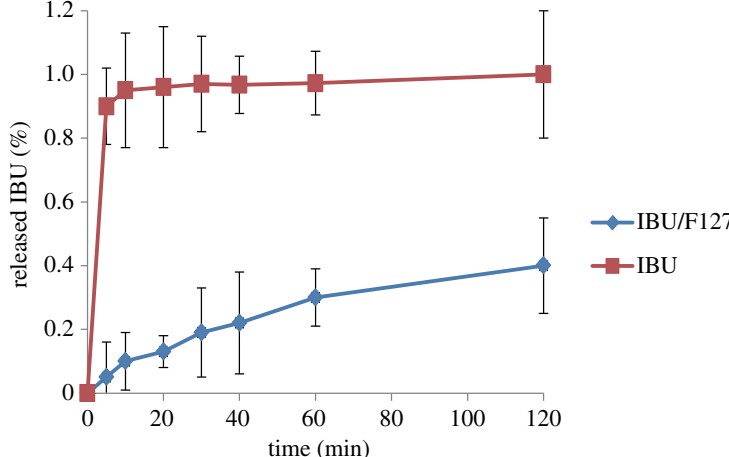

**Figure 7.** Release curve of ibuprofen on mesoporous silicate molecular sieves by aerosol-assisted method.

mixed well. The TEOS was added after the template was dissolved. Then the mix was hydrolyzed at room temperature for 4 h. The aerosol spray was generated by the mixed liquid through the aerosol generator and the dried at 300°C in a tube heating furnace. In this process, as the volatile solvent evaporated, the non-volatile matter was self-assembled into the mesoporous structure to obtain a white powder. The white powder was dried at 100°C overnight and baked at 540°C for 6 h to remove the template to obtain the final product. The product was labelled as F127.

## 3.2. Characterization

XRD analysis was performed in a FEI Tecnai G2 F30 X-ray powder diffractometer. TEM images were obtained on the Tecnai G220 Stwin-type electron microscope from FEI. The acceleration voltage was 200 kV, and the sample was ultrasonically oscillated in absolute ethanol before the test. SEM images were recorded using a Helios with an acceleration voltage of 10 kV. The $N_2$ adsorption–desorption isotherm ($N_2$-TPD) of the sample was measured using the American-based Quantachome company's Autosorb-1 physical adsorption instrument. The specific surface area was calculated using the modified Brunner Emmet Teller method. The adsorption pressure corresponding to the pore volume was $P/P_0 = 0.99$. Calculation of pore size distribution of mesoporous molecular sieves was by the Barret Joyner Halenda method.

## 3.3. The loading of ibuprofen on molecular sieves

Liquid ultraviolet spectrophotometry was used for quantitative analysis of drug molecules. The IBU was added into an appropriate amount of ethanol (35 mg g$^{-1}$). The mix was magnetically stirred until

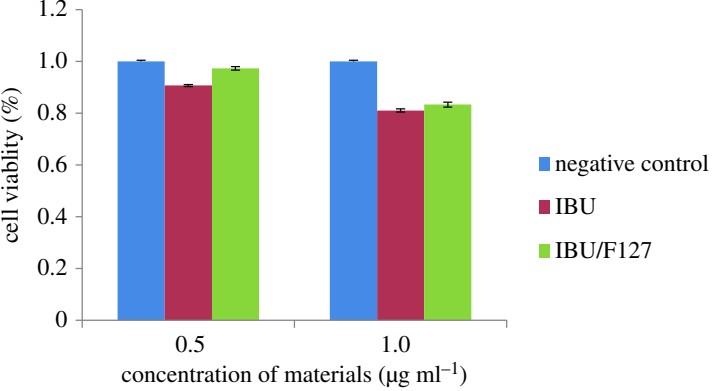

**Figure 8.** The bio-safety of ibuprofen on mesoporous silicate molecular sieves by the aerosol-assisted method.

dissolved. An amount of molecular sieves were added. Samples were taken at regular intervals and diluted appropriately for ultraviolet testing. Two hundred and seventy-three nanometres was the wavelength of incident light. According to the IBU standard solution, the standard curve was measured as $C = 0.613A$-0.009 ($r = 0.9998$, $n = 8$). The loading of IBU per unit mass molecular sieve was calculated by the differential subtraction method. The specific calculation formula was as follows. $L$ was the loading of IBU in the molecular sieve, $m_0$ was the initial mass of IBU in the solution, $c_n$ was the concentration of the residual drug solution, $M_0$ was the initial mass of the solution, $V$ was the volume of the sample solution (1 ml) and $M$ was the mass added to the molecular sieve:

$$L = \frac{m_0 - c_n \times (M_0 - V \sum_{i}^{n-1} c_i)}{M}.$$

After the loading process was completed, the sample was filtered by vacuum suction filtration, washed by ethanol and dried at room temperature to obtain a molecular sieve loaded with IBU. The sample was labelled as IBU/F127.

## 3.4. The release of ibuprofen on molecular sieves

The simulated intestinal fluid (pH = 7.4 phosphate-buffered solution) was used as the release system solvent. The IBU was added into an appropriate amount of phosphate-buffered solution. The mix was magnetically stirred. Samples were taken at regular intervals and diluted appropriately for ultraviolet testing. Derivative spectroscopy was used to process the data and the regression equation of the derivative spectrum of IBU in phosphate buffer was dl(A) = 0.762C-0.017 ($r = 0.9997$, $n = 8$). IBU/F127 was pressed into tablets and put into the release solution. The sample was stirred at room temperature and taken at regular intervals. The same amount of fresh medium was replenished. The supernatant taken was diluted appropriately for ultraviolet testing. The amount of IBU released per unit molecular sieve was as follows. $R$ was the amount of IBU released per drug-loaded molecular sieve, $c_n$ was the measured IBU concentration in the released $V_0$ solution, $V_0$ was the total volume of the release medium (500 ml), $V$ is the sampling volume (5 ml) and $M$ is the mass of the load tablet:

$$R = \frac{c_n V_0 + V \sum_{i}^{n-1} c_i}{M}.$$

## 3.5. The bio-safety of materials (cell counting Kit-8)

The cell counting Kit-8(CCK-8) assay was used to determine the bio-safety of materials. Cell lines HIEC-6 were maintained in RPMI 1640 (Life Technologies) with 10% fetal bovine serum (Sigma) in a humidified atmosphere at 37°C with 5% $CO_2$. Briefly, cells were exposed to different concentrations of materials, and 48 h after treatment, cells were processed for the CCK-8 assay according to the standard methods.

# 4. Conclusion

In conclusion, we present a simple and efficient aerosol-assisted method to prepare novel mesoporous silicate molecular sieves. The investigation results demonstrate that mesoporous silicate molecular sieves by the aerosol-assisted method are spherical with a core–shell structure. The morphology of most synthesized samples is spherical with a particle size of about 200 nm. The layer of annular pores is arranged in parallel on the outside and the three-dimensional short worm-like pore structure is arranged inside the spherical particles. The drug IBU was applied as a model drug to investigate the drug release behaviour by ultraviolet spectrophotometry measurements. As a drug carrier, it has good structural stability and can achieve drug controlled release which is expected. It exhibits safety to a certain degree. Therefore, the aerosol-assisted synthesis method provides a new idea for the synthesis of sustained-release drug carriers.

Data accessibility. Raw data for the figures is included as the electronic supplementary material. Our data are deposited at Dryad. https://doi.org/10.5061/dryad.tmpg4f4w4 [43].

Authors' contributions. Y.W. conceived and designed the experiments; L.H. performed the experiments; X.L. analysed the data; L.H. wrote the paper.

Competing interests. The authors declare no conflict of interest.

Funding. This study is financially supported by the grant from Research Startup Fund of China Medical University (XZR20160035) and China Postdoctoral Science Foundation (2019M661160).

Acknowledgements. We thank the device support of Laboratory of China Medical University.

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
