## [Reviewer comments · Royal Society Open Science]

Review History

RSOS-200650.R0 (Original submission)

Review form: Reviewer 1

Is the manuscript scientifically sound in its present form?

Yes

Are the interpretations and conclusions justified by the results?

Yes

Is the language acceptable?

Yes

Do you have any ethical concerns with this paper?

No

Have you any concerns about statistical analyses in this paper?

No

Recommendation?

Accept with minor revision (please list in comments)

Comments to the Author(s)

In this article, the authors report the aerosol-assisted synthesis of spherical mesoporous silicate molecular sieves with sustained ibuprofen release performance. The presented gas-phase process may address the limitations of conventional wet chemistry-based routes.

The work lies well in the scope of the Royal Society Open Science. The results are significant and might attract broad interest. I thus recommend it for publication after minor revisions addressed as follows.

1. The authors should remove the word “novel” from the title because there are some publications that used a similar method (Materials Today: Proceedings 3 (2016) 646 – 651).

2. There are several spelling faults in the manuscript, which need to be corrected to improve the quality of the paper. Some examples are listed in bellow:

Page 2, in the title: by Aerosol-assisted method=> by aerosol-assisted method

Page 6, line 27: Fig. 3 => Figure 3

Page 7, line 37: Fig. 5A => Figure 5A

Page 8, line 45: 35 mg/g-1=> 35 mg/g

Page 9, in the horizontal axis of Figure 6: Time (h)

3. On page 3, line 33, the authors should cite a reference for Vallet Regi.

Review form: Reviewer 2

Is the manuscript scientifically sound in its present form?

Yes

Are the interpretations and conclusions justified by the results?

Yes

Is the language acceptable?

Yes

Do you have any ethical concerns with this paper?

No

Have you any concerns about statistical analyses in this paper?

No

Recommendation?

Accept with minor revision (please list in comments)

Comments to the Author(s)

Manuscript ID: RSOS-200650

This manuscript from Liang Hao and coworkers is reporting a good work about the novel mesoporous silicate molecular sieves were synthesized with polyether F127 as the template by aerosol-assisted method for loading and release of ibuprofen. The synthesized samples were characterized adequately. The results suggested the aerosol-assisted synthesis method provides a new idea for the synthesis of sustained-release drug carriers. This study is an interesting work for design novel mesoporous silicate molecular sieves for drug delivery. I would like to recommend the publication of this work after revision. However, the authors should solve the following issues.

The issues in the manuscript are listed as below:

1. Introduction, the design of drug delivery systems based on mesoporous silicate had been investigated wildly in the field of biomedical, and some relative works must be cited to enrich the background of this paper. Please see references (Advanced Functional Materials 27 (2017) 1605985; Small 13 (2017) 1700623; Advanced Functional Materials 27 (2017) 1704135; Nanoscale 9 (2017) 17063-17073; Journal of Colloid and Interface Science 525 (2018) 1-10).
2. The FT-IR and XPS characterization of mesoporous silicate molecular sieves could be presented, if possible? Please see (Advanced Science 5 (2018) 1800510; Biomaterials Science 6 (2018) 1084-1098).
3. The of Figure 6 should be checked and corrected.
4. How many times of ibuprofen release from mesoporous silicate molecular sieves? The standard deviation should be shown.
5. The bio-safety of mesoporous silicate molecular sieves could be evaluated, if possible?
6. Overall the paper is easily read, but there are still some minor problems in language and spelling. A throughout checking is recommended.

The manuscript could be considered for publication only after revision. Moreover, I am glad to review a revision of this manuscript if necessary.

Decision letter (RSOS-200650.R0)

Dear Dr Hao:

Title: Synthesis of novel mesoporous silicate molecular sieves by Aerosol-assisted method for loading and release of drug
Manuscript ID: RSOS-200650

Thank you for submitting the above manuscript to Royal Society Open Science. On behalf of the Editors and the Royal Society of Chemistry, I am pleased to inform you that your manuscript will be accepted for publication in Royal Society Open Science subject to minor revision in accordance with the referee suggestions. Please find the reviewers' comments at the end of this email. I apologise that this has taken longer than usual.

The reviewers and handling editors have recommended publication, but also suggest some minor revisions to your manuscript. Therefore, I invite you to respond to the comments and revise your manuscript.

Because the schedule for publication is very tight, it is a condition of publication that you submit the revised version of your manuscript before 13-Aug-2020. Please note that the revision deadline will expire at 00.00am on this date. If you do not think you will be able to meet this date please let me know immediately.

When submitting your revised manuscript, you will be able to respond to the comments made by the referees and upload a file "Response to Referees" in "Section 6 - File Upload". You can use this

to document any changes you make to the original manuscript. In order to expedite the processing of the revised manuscript, please be as specific as possible in your response to the referees.

Kind regards,
 Dr Laura Smith
 Publishing Editor, Journals
 Royal Society of Chemistry
 Thomas Graham House
 Science Park, Milton Road
 Cambridge, CB4 0WF
 Royal Society Open Science - Chemistry Editorial Office

On behalf of the Subject Editor Professor Anthony Stace and the Associate Editor Dr Dattatray Late.

RSC Subject Editor:
 Comments to the Author:
 (There are no comments.)

RSC Associate Editor:
 Comments to the Author:
 The aerosol-assisted method were reported for the preparation of mesoporous silicate molecular sieves with detail morphological and structural characterization details.

Reviewer comments to Author:

Reviewer: 1

Comments to the Author(s)

In this article, the authors report the aerosol-assisted synthesis of spherical mesoporous silicate molecular sieves with sustained ibuprofen release performance. The presented gas-phase process may address the limitations of conventional wet chemistry-based routes.

The work lies well in the scope of the Royal Society Open Science. The results are significant and might attract broad interest. I thus recommend it for publication after minor revisions addressed as follows.

1. The authors should remove the word “novel” from the title because there are some publications that used a similar method (*Materials Today: Proceedings* 3 (2016) 646 – 651).

2. There are several spelling faults in the manuscript, which need to be corrected to improve the quality of the paper. Some examples are listed in bellow:

Page 2, in the title: by Aerosol-assisted method=> by aerosol-assisted method

Page 6, line 27: Fig. 3 => Figure 3

Page 7, line 37: Fig. 5A => Figure 5A

Page 8, line 45: 35 mg/g-1=> 35 mg/g

Page 9, in the horizontal axis of Figure 6: Time (h)

3. On page 3, line 33, the authors should cite a reference for Vallet Regi.

Reviewer: 2

Comments to the Author(s)

Manuscript ID: RSOS-200650

This manuscript from Liang Hao and coworkers is reporting a good work about the novel mesoporous silicate molecular sieves were synthesized with polyether F127 as the template by aerosol-assisted method for loading and release of ibuprofen. The synthesized samples were characterized adequately. The results suggested the aerosol-assisted synthesis method provides a new idea for the synthesis of sustained-release drug carriers. This study is an interesting work for design novel mesoporous silicate molecular sieves for drug delivery. I would like to recommend the publication of this work after revision. However, the authors should solve the following issues.

The issues in the manuscript are listed as below:

1. Introduction, the design of drug delivery systems based on mesoporous silicate had been investigated wildly in the field of biomedical, and some relative works must be cited to enrich the background of this paper. Please see references (*Advanced Functional Materials* 27 (2017) 1605985; *Small* 13 (2017) 1700623; *Advanced Functional Materials* 27 (2017) 1704135; *Nanoscale* 9 (2017) 17063-17073; *Journal of Colloid and Interface Science* 525 (2018) 1-10).

2. The FT-IR and XPS characterization of mesoporous silicate molecular sieves could be presented, if possible? Please see (*Advanced Science* 5 (2018) 1800510; *Biomaterials Science* 6 (2018) 1084-1098).

3. The of Figure 6 should be checked and corrected.

4. How many times of ibuprofen release from mesoporous silicate molecular sieves? The standard deviation should be shown.

5. The bio-safety of mesoporous silicate molecular sieves could be evaluated, if possible?

6. Overall the paper is easily read, but there are still some minor problems in language and spelling. A throughout checking is recommended.

The manuscript could be considered for publication only after revision. Moreover, I am glad to review a revision of this manuscript if necessary.

Author's Response to Decision Letter for (RSOS-200650.R0)

See Appendix A.

RSOS-200650.R1 (Revision)

Review form: Reviewer 1

Is the manuscript scientifically sound in its present form?

Yes

Are the interpretations and conclusions justified by the results?

Yes

Is the language acceptable?

Yes

Do you have any ethical concerns with this paper?

No

Have you any concerns about statistical analyses in this paper?

No

Recommendation?

Accept as is

Comments to the Author(s)

The authors have done a great job addressing all reviewer comments, and I have no further suggestions. I believe the paper is acceptable for publication in the Royal Society Open Science.

Review form: Reviewer 2

Is the manuscript scientifically sound in its present form?

Yes

Are the interpretations and conclusions justified by the results?

Yes

Is the language acceptable?

Yes

Do you have any ethical concerns with this paper?

No

Have you any concerns about statistical analyses in this paper?

No

Recommendation?

Accept as is

Comments to the Author(s)

The revised manuscript has been improved greatly and it could be accepted for publication.

Decision letter (RSOS-200650.R1)

Dear Dr Hao:

Title: Synthesis of novel mesoporous silicate molecular sieves by Aerosol-assisted method for loading and release of drug

Manuscript ID: RSOS-200650.R1

It is a pleasure to accept your manuscript in its current form for publication in Royal Society Open Science. The chemistry content of Royal Society Open Science is published in collaboration with the Royal Society of Chemistry.

On behalf of the Subject Editor Professor Anthony Stace and the Associate Editor Dr Dattatray Late.

RSC Associate Editor:
Comments to the Author:
Accept as is

RSC Subject Editor:
Comments to the Author:
(There are no comments.)

Reviewer(s)' Comments to Author:

Reviewer: 1

Comments to the Author(s)

The authors have done a great job addressing all reviewer comments, and I have no further suggestions. I believe the paper is acceptable for publication in the Royal Society Open Science.

Reviewer: 2

Comments to the Author(s)

The revised manuscript has been improved greatly and it could be accepted for publication.

Appendix A

Aug 8, 2020

Professor Manfred Bochmann
Editor
Royal Society Open Science
Dear Editor & Prof. Laura Smith,

Please kindly find the submitted revised manuscript No. RSOS-200650 “Synthesis of novel mesoporous silicate molecular sieves by Aerosol-assisted method for loading and release of drug”, contributed from Liang Hao, Xiaojia Li, Yang Wang.

We have revised the manuscript according to your suggestions and the comments from reviewers, especially the format and language of the manuscript (the revised parts are marked in red). The followings are the comments from the two reviewers and our one-to-one responses.

Comments from reviewer 1:

1. The authors should remove the word “novel” from the title because there are some publications that used a similar method (Materials Today: Proceedings 3 (2016) 646 – 651).

Answer:

We followed the advice and changed the title to “**Synthesis of mesoporous silicate molecular sieves by Aerosol-assisted method for loading and release of drug**”.

2. There are several spelling faults in the manuscript, which need to be corrected to improve the quality of the paper. Some examples are listed in bellow:

Page 2, in the title: by Aerosol-assisted method=> by aerosol-assisted method

Page 6, line 27: Fig. 3 => Figure 3

Page 7, line 37: Fig. 5A => Figure 5A

Page 8, line 45: 35 mg/g-1=> 35 mg/g

Page 9, in the horizontal axis of Figure 6: Time (h)

Answer:

Thanks for the referee’s advices. All these suggestions are adopted, and the corresponding paragraphs in the text have been corrected.

3. On page 3, line 33, the authors should cite a reference for Vallet Regi.

Answer:

We followed the referee’s kind suggestion and added the reference in the revised manuscript.

Comments from reviewer 2:

1. Introduction, the design of drug delivery systems based on mesoporous silicate had been investigated widely in the field of biomedical, and some relative works must be cited to enrich the background of this paper. Please see references (Advanced Functional Materials 27 (2017) 1605985; Small 13 (2017) 1700623; Advanced Functional Materials 27 (2017) 1704135; Nanoscale 9 (2017) 17063-17073; Journal of Colloid and Interface Science 525 (2018) 1-10).

Answer:

To follow the referee's suggestion and the relative references, we revised the introduction and cited the references.

2. The FT-IR and XPS characterization of mesoporous silicate molecular sieves could be presented, if possible? Please see (Advanced Science 5 (2018) 1800510; Biomaterials Science 6 (2018) 1084-1098).

Answer:

Following the reviewer's suggestion, we regret very much to find that we tried FT-IR was shown as follow, Si-O bond were shown around 1100 cm^{-1} , ibuprofen were shown around 1700 cm^{-1} , but there are no signals about ibuprofen loading on the sieves or the peak overlaps with other peaks due to the equipment. The loaded ibuprofen may decompose due to the samples were stored for several months. We didn't use any special elements during the preparation of molecular sieves, so the measurement of XPS wasn't done. It seems that the loaded ibuprofen decomposed owing to the sample store for a long time. Unfortunately, we can't supplement XPS data.

3. The of Figure 6 should be checked and corrected.

Answer:

Thanks for the referee's advices. we supplement the Figure 6 with standard deviation in the manuscript.

4. How many times of ibuprofen release from mesoporous silicate molecular sieves?

The standard deviation should be shown.

Answer:

Each releasing experiment was done three times. Follow the referee's suggest, we supplement the Figure 6 with standard deviation in the manuscript. ◦

Figure 6

5. The bio-safety of mesoporous silicate molecular sieves could be evaluated, if possible?

Answer:

To evaluate the bio-safety of mesoporous silicate molecular sieves, HIEC-6 cell was adopt for cytotoxicity test. And the results were shown in Figure 8.

6. Overall the paper is easily read, but there are still some minor problems in language and spelling. A throughout checking is recommended.

Answer:

Follow the referee's advice, we have checked the whole manuscript with great care.

I look forward to hearing from you at your earliest convenience. Thank you very much in advance for your assistance in this matter.

Thanks for your attention.

Sincerely yours,

Yang WANG

School of Fundamental Sciences

China Medical University

Shenyang 110122, R. P. China

Tel.: +86-18909833907

E-mail address: 563484708@qq.com